# When One Moment Isn't Enough: Multi-Moment Retrieval with Cross-Moment Interactions

**Zhuo Cao**[1*], **Heming Du**[1*], **Bingqing Zhang**[1], **Xin Yu**[1], **Xue Li**[1†], **Sen Wang**[1]

[1] The University of Queensland, Australia
{william.cao, heming.du, bingqing.zhang, xin.yu}@uq.edu.au
xueli@eesc.uq.edu.au, sen.wang@uq.edu.au

## Abstract

Existing Moment retrieval (MR) methods focus on Single-Moment Retrieval (SMR). However, one query can correspond to multiple relevant moments in real-world applications. This makes the existing datasets and methods insufficient for video temporal grounding. By revisiting the gap between current MR tasks and real-world applications, we introduce a high-quality datasets called QVHighlights Multi-Moment Dataset (QV-M$^2$), along with new evaluation metrics tailored for multi-moment retrieval (MMR). QV-M$^2$ consists of 2,212 annotations covering 6,384 video segments. Building on existing efforts in MMR, we propose a framework called FlashMMR. Specifically, we propose a Multi-moment Post-verification module to refine the moment boundaries. We introduce constrained temporal adjustment and subsequently leverage a verification module to re-evaluate the candidate segments. Through this sophisticated filtering pipeline, low-confidence proposals are pruned, and robust multi-moment alignment is achieved. We retrain and evaluate 6 existing MR methods on QV-M$^2$ and QVHighlights under both SMR and MMR settings. Results show that QV-M$^2$ serves as an effective benchmark for training and evaluating MMR models, while FlashMMR provides a strong baseline. Specifically, on QV-M$^2$, it achieves improvements over prior SOTA method by 3.00% on G-mAP, 2.70% on mAP@3+tgt, and 2.56% on mR@3. The proposed benchmark and method establish a foundation for advancing research in more realistic and challenging video temporal grounding scenarios. Code is released at https://github.com/Zhuo-Cao/QV-M2.

## 1 Introduction

Understanding how natural language relates to visual events in videos is a core problem in video-language research [33, 43, 42, 13, 37]. One representative task, Moment Retrieval (MR), aims to retrieve relevant temporal segments given a natural language query. Most existing MR methods [39, 17, 6, 12] operate under the Single-Moment Retrieval (SMR) paradigm, assuming that each query corresponds to exactly one relevant moment within a video. However, this assumption oversimplifies real-world scenarios, where a single query often aligns with multiple non-overlapping moments. For example, in instructional videos, a query such as "cutting vegetables" may correspond to several separate instances of chopping different ingredients throughout the video. Similarly, in sports broadcasts, "successful three-point shots" may occur multiple times within a single match.

Despite the prevalence of such multi-moment scenarios, existing MR methods [12, 20, 18, 21] remain inherently limited to retrieving only the most relevant single segment, disregarding other

---

*Equal Contribution
†Corresponding Authors

39th Conference on Neural Information Processing Systems (NeurIPS 2025).

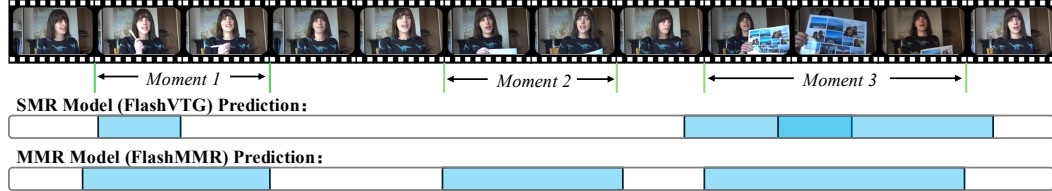

Query: *"A woman with short brown hair and curved bangs holding an object in her hand."*

Figure 1: **Prediction in SMR vs. MMR.** In multi-moment retrieval, SMR optimizes for the most probable single moment, often disregarding other valid segments. In contrast, MMR encourages comprehensive retrieval by identifying all semantically relevant moments, aligning better with real-world video understanding.

valid moments. This constraint severely limits their applicability to comprehensive video-language understanding. Addressing this fundamental gap between current MR methodologies and real-world applications requires a paradigm shift from SMR to Multi-Moment Retrieval (MMR).

By revisiting current video temporal grounding research [2, 18, 20], we found that a major limitation is the lack of standardized datasets and evaluation metrics tailored for MMR. To support rigorous benchmarking and further research in this area, we introduce QV-M$^2$ (QVHighlights [12] Multi-Moment Dataset), an enhanced dataset based on QVHighlights. QV-M$^2$ contains 2,212 high-quality human-annotated queries, covering 6,384 annotated temporal segments across diverse video scenarios. Unlike previous MR datasets, which primarily support SMR evaluation, QV-M$^2$ explicitly accounts for queries with multiple relevant moments, making it the first fully human-annotated dataset dedicated to MMR benchmarking. Beyond dataset contributions, we also propose new evaluation metrics that extend standard mean Average Precision (mAP) and Intersection-over-Union (IoU) to measure MMR performance. These metrics provide a comprehensive evaluation considering both the SMR and the MMR settings, ensuring that future MMR research aligns with real-world requirements.

To further advance MMR, we introduce FlashMMR, a novel framework explicitly designed for MMR. A key challenge in MMR is to ensure that the retrieved moments precisely include all relevant moments and exclude false positives. To achieve this, we propose a Multi-Moment Post-Verification module, which refines moment boundaries through a constrained temporal adjustment strategy and further verifies retrieved moments using a semantic consistency-based re-evaluation process. This module filters out low-confidence proposals and enhances alignment with query semantics. Through this structured refinement pipeline, FlashMMR effectively reduces the presence of redundant or irrelevant moment predictions while maximizing the recall for multiple relevant instances. As show in Figure 1, the objective shifts from retrieving the single most relevant moment to identifying as many relevant and semantically consistent moments as possible.

To validate the effectiveness of FlashMMR, we benchmark 6 open-source MR models under both SMR and MMR settings on QV-M$^2$ and QVHighlights. Experimental results show that the new dataset enhances the performance of all methods on both SMR and MMR task. Our comprehensive comparisons reveal that while existing methods perform well in single-moment retrieval, they struggle significantly in multi-moment scenarios due to their inherent architectural limitations. In contrast, FlashMMR consistently outperforms prior approaches across all MMR metrics, demonstrating its effectiveness, with a 3.00% improvement in G-mAP, 2.70% in mAP@3+tgt, and 2.56% in mR@3 on QV-M$^2$. These findings underscore the necessity of dedicated MMR frameworks to advance video language understanding. FlashMMR not only enhances the performance of MMR task, but also establishes a new benchmark for future research in MMR.

In summary, Our contributions are as follows:

1. We introduce FlashMMR, a novel Multi-Moment Retrieval (MMR) framework, which incorporates a Multi-Moment Post-Verification module to refine candidate segments by enforcing temporal consistency across related moments.

2. We propose QV-M$^2$, the first fully human-annotated MMR dataset, designed to facilitate benchmarking and model development for MMR.

3. We develop a comprehensive suite of MMR evaluation metrics that extend traditional MR evaluation protocols. These metrics jointly assess retrieval accuracy and temporal coverage, offering a fine-grained evaluation of model performance on moment retrieval.

Table 1: Comparison with existing moment retrieval datasets.

| Dataset | Domain | Avg #moment per query | Avg Query Len | Avg ratio Moment/Video | #Moments / #Videos | Fully Human-Annotated |
|---|---|---|---|---|---|---|
| DiDeMo [1] | Flickr | 1 | 8.0 | 22.2% | 41.2K / 10.6K | ✓ |
| ANetCaptions [10] | Activity | 1 | 14.8 | 30.8% | 72K / 15K | ✓ |
| CharadesSTA [5] | Activity | 1 | 7.2 | 26.5% | 16.1K / 6.7K | ✗ |
| TVR [11] | TV show | 1 | 13.4 | 12.0% | 109K / 21.8K | ✓ |
| TACoS [27] | Cooking | 1 | 27.9 | 8.4% | 18K / 0.1K | ✓ |
| YouCook2 [47] | Cooking | 1 | 8.7 | 1.9% | 13.8K / 2K | ✓ |
| COIN [38] | Open | 1 | 4.9 | 9.8% | 46.3K / 11.8K | ✓ |
| HiREST [40] | Open | 1 | 4.2 | 55.0% | 2.4K / 0.5K | ✓ |
| NExT-VMR [25] | Open | 1.5 | – | – | 229.5K / 9K | ✗ |
| QVHighlights [12] | Vlog/News | 1.8 | 11.3 | 16.4% | 18.5K / 10.2K | ✓ |
| **QV-M²** (Ours) | Vlog/News | 2.9 | 12.0 | 25.5% | 6.4K / 1.3K | ✓ |

By providing a strong benchmark and a new dataset, our work lays the foundation for future research on MMR systems, paving the way for more realistic video understanding in complex environments.

## 2 Related Work

### 2.1 Single-Moment Retrieval

Single-Moment Retrieval (SMR) focuses on the task of localizing a single relevant temporal segment within a video based on a natural language query. Given the query, the model predicts the start and end timestamps of the most relevant moment. Existing SMR methods [29, 7] can be broadly categorized into proposal-based and proposal-free approaches.

**Proposal-Based Methods.** Proposal-based approaches decompose the retrieval process into two stages: candidate moment generation followed by matching and ranking. These methods can be further classified based on how they generate candidate segments: Sliding Window Approaches [39, 17, 6, 1, 5] systematically segment videos into overlapping windows and evaluate their relevance to the query. Anchor-Based Approaches [46, 39, 32, 44] define a set of predefined anchors across the video and refine the most promising candidates. Proposal-Generated Approaches [45, 36, 34, 30, 15] utilize deep neural networks to generate adaptive moment proposals.

**Proposal-Free Methods.** Proposal-free approaches directly predict the start and end timestamps using sequence regression techniques. Instead of evaluating pre-defined candidate segments, these methods formulate moment retrieval as an end-to-end sequence prediction problem [16, 19, 41, 22, 12, 28]. Recent transformer-based models, such as Moment-DETR [12], adopt a set-based prediction paradigm to eliminate the need for non-maximum suppression (NMS) and other post-processing steps.

While proposal-free methods demonstrate higher efficiency and flexibility, they often struggle with multi-moment scenarios, as they are inherently designed to retrieve only a single moment per query.

### 2.2 Multi-Moment Retrieval

Real-world video content often contains multiple non-overlapping moments that are semantically relevant to the query. Current SMR models address this by selecting the highest IoU moment as the ground truth, ignoring other valid segments [12]. This simplification results in suboptimal retrieval performance when multiple events contribute to the semantics of the query. To address these shortcomings, recent studies have explored Multi-Moment Retrieval (MMR), where a query can be mapped to multiple relevant moments [25, 8].

**Early Efforts in MMR.** Early works attempted to adapt SMR models to MMR by modifying retrieval pipelines: Otani *et al.* [24] identified false negatives in SMR training. Liu *et al.* [17] extended proposal-based MR models to allow multiple moment predictions per query, though lacking robust mechanisms for handling dependencies among moments.

**Recent Advances in MMR.** Recent works have introduced dedicated architectures and datasets for MMR: SFABD [8] refines candidate retrieval by eliminating false negatives and improving

alignment with query semantics. Concurrent work by Qin *et al.* [25] extends moment retrieval beyond single-moment assumptions, introducing NExT-VMR, a dataset designed to support multi-moment and no-moment retrieval. However, its design and evaluation remain closely aligned with previous benchmarks, without targeted optimizations for the unique challenges of MMR. Additionally, the dataset is not yet publicly available.

Despite recent efforts in multi-moment retrieval (MMR), a high-quality dataset built with standardized methodology is still lacking. We address this gap by introducing QV-M$^2$, a densely annotated, high-quality MMR benchmark, together with FlashMMR, which establishes a new baseline for the multi-moment retrieval task.

## 3 QV-M$^2$ Dataset

### 3.1 Video Collection and Annotation

**Video Collection.** To better reflect real-world MMR scenarios and facilitate comparison with existing SMR tasks, building on existing moment retrieval datasets is a natural and effective choice for developing MMR benchmarks. Among them, QVHighlights [12] stands out as one of the most widely adopted datasets. Composed of unedited or minimally edited videos with naturally rich content, it serves as an ideal foundation for extending to MMR while enabling a seamless transition and comparison with SMR.

Our dataset retains the original QVHighlights videos and adds new annotations for MMR. The videos, sourced from YouTube, include lifestyle vlogs and news footage, covering diverse scenarios such as travel, social activities, natural disasters, and protests. They vary in perspective (e.g., first-person, third-person) and range from 5 to 30 minutes in length, offering both diversity and annotation feasibility. This ensures the dataset captures real-world complexity and better supports the MMR task.

**Video Annotation.** We design a manual annotation process for the one-to-many nature of MMR. It improves diversity and coverage while ensuring high quality. We define a set of annotation guidelines to standardize the process, including: (i) create *detailed queries* that precisely capture actors, actions, and contexts; (ii) include *context-dependent queries* that require knowledge of temporal relationships within the video; and (iii) design *negative (inverse) queries* to mark segments where a specified action or event does *not* occur.

Each query must be matched with one or multiple video segments. To facilitate this, we develop an annotation interface that allows annotators to watch each video, formulate the queries, and assign start and end times for relevant segments. Details of the annotation interface are provided in supplementary material. For quality control, after every 100 videos are annotated by the primary annotator, we randomly sample 5% of these videos for re-checking by an additional annotator. If the temporal boundaries identified by the two annotators overlap by less than 90%, the batch is re-annotated by a third annotator. This process ensures annotation consistency and captures the complexity of MMR. License and data usage details are provided in the supplementary material.

### 3.2 Dataset Statistics

Our dataset, QV-M², consists of 2,212 new queries associated with 1,341 videos, covering a total of 6,384 annotated temporal moments. As shown in Table 1, compared to existing moment retrieval datasets, QV-M² is distinct in its MMR setting, where each query is linked to an average of 2.9 moments, significantly surpassing the typical single-moment assumption in prior datasets.

To analyze the temporal properties of annotated moments, we present the distribution of moment lengths and location in Figure 2(a) and (c). As shown in (a), the majority of temporal windows fall within the 2 to 20-second range, with a notable 1,263 instances (19.8%) extending beyond 20 seconds, highlighting the diversity in temporal granularity.

Figure 2(c) compares the ground-truth moment boundaries of QV-M$^2$ and Charades-STA [5] (start time on the x-axis and end time on the y-axis; both axes normalized by video duration), and reveals that QV-M$^2$ annotations are distributed more uniformly throughout videos. Lexical analysis of QV-M² (Figure 2(b)) reveals a rich vocabulary of commonly occurring nouns and verbs, emphasizing key concepts in human interactions, fashion, and daily activities. Frequent words such as *woman, man,*

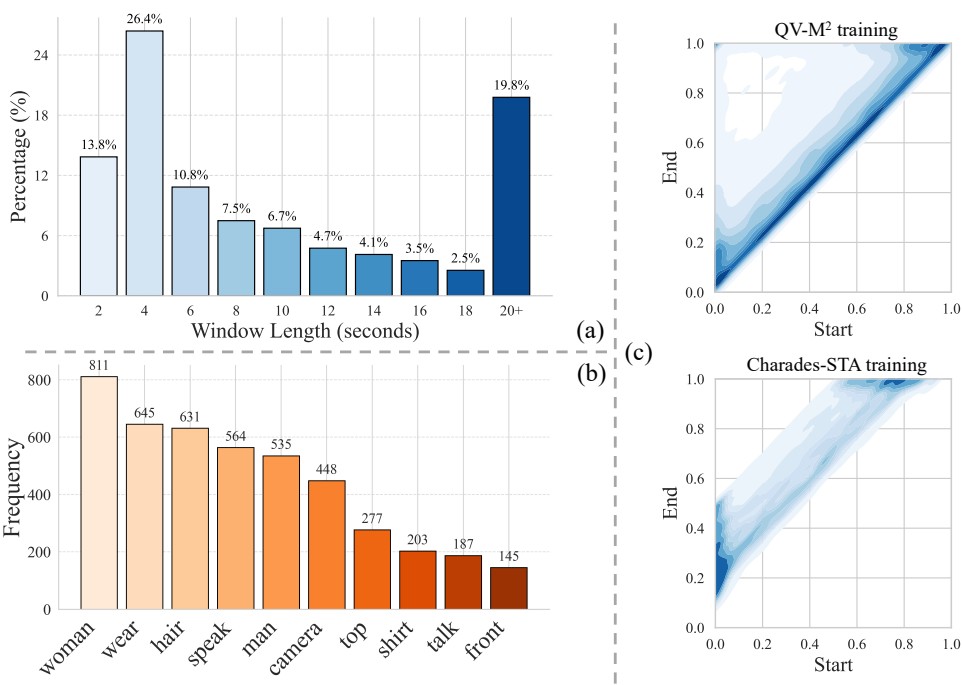

Figure 2: Dataset statistics for QV-M$^2$. (a) and (c) show the distributions of moment lengths and temporal locations, respectively; (b) reports the top-10 most frequent words in the annotations.

*speak, wear,* and *scene* suggest a broad coverage of people-centric activities, aligning well with the dataset's focus on vlog and news content.

## 3.3 Evaluation Metrics For MMR

We develop new evaluation metrics by smoothly adapting SMR metrics to handle multiple moments. Specifically, we label a prediction as a true positive if its Intersection-over-Union (IoU) with any unmatched ground truth meets a threshold (*e.g.*, 0.5); otherwise, it is deemed a false positive.

**Generalized mAP.** The generalized mean Average Precision (G-mAP) is computed by averaging the AP scores over multiple IoU thresholds:

$$\text{G-mAP} = \frac{1}{|\mathcal{T}|} \sum_{\tau \in \mathcal{T}} \text{AP}(\tau),$$

where $\mathcal{T}$ denotes the set of IoU thresholds (*e.g.*, $\{0.5, 0.55, \ldots, 0.9\}$), and $\text{AP}(\tau)$ represents the average precision computed at threshold $\tau$.

To further capture performance nuances, we categorize queries by their number of ground-truth moments and report mAP under each category (*e.g.*, $mAP@1\_tgt$, $mAP@2\_tgt$, $mAP@3+\_tgt$). Averaging these scores over multiple IoU thresholds also yields the G-mAP, ensuring a robust, unified metric that can evaluate both single- and multi-target scenarios.

**Mean IoU@$k$.** The mean Intersection-over-Union at top-$k$ predictions is defined as:

$$\text{mIoU}@k = \frac{1}{|\mathcal{Q}|} \sum_{q \in \mathcal{Q}} \frac{1}{k} \sum_{i=1}^{k} \max_{\text{gt} \in \mathcal{G}(q)} \text{IoU}(\text{pred}_i, \text{gt}),$$

where $k \in \{1, 2, 3\}$ denotes the rank, $\mathcal{Q}$ is the set of all queries, and $\mathcal{G}(q)$ represents the ground-truth moments associated with query $q$. The function $\text{IoU}(\text{pred}_i, \text{gt})$ computes the Intersection-over-Union between the $i$-th prediction and one of the ground-truth moment. It is worth noting that Mean IoU@$k$ is computed only on queries with at least $k$ ground-truth moments.

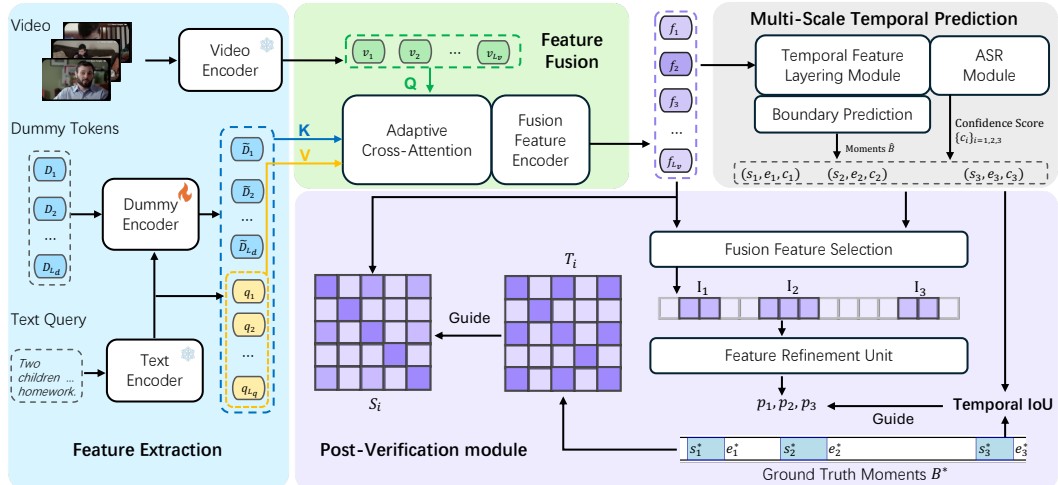

Figure 3: **Overview of the FlashMMR Framework.** In Feature Extraction, video and query features are extracted via frozen encoders. The textual feature, combined with encoded dummy tokens, forms the *key*, while the video feature serves as the *query* and the text feature as the *value* in Feature Fusion Module. This produces fused features $\{f_i\}_{i=1}^{L_v}$ (in purple), where color intensity indicates semantic relevance. During inference, the fused features are directly passed into the Multi-Scale Temporal Prediction Module to generate the final prediction $\{(s_i, e_i, c_i)\}_{i=1}^3$. During training, the Post-Verification Module further refines the initial prediction. Specifically, the fused features are aligned with the prediction to obtain a refined confidence score $p_i$ and a self-similarity matrix $S_i$, both of which are supervised using the ground truth moments.

**Mean Recall@$k$.** The recall at top-$k$ predictions is defined as:

$$\text{mR@}k = \frac{1}{|\mathcal{Q}|} \sum_{q \in \mathcal{Q}} \frac{1}{|\mathcal{G}(q)|} \sum_{\text{gt} \in \mathcal{G}(q)} \mathbf{1} \left[ \max_{i \leq k} \text{IoU}(\text{pred}_i, \text{gt}) \geq \tau \right].$$

where $k \in \{1, 2, 3\}$ denotes the rank, $\mathbf{1}[\cdot]$ is the indicator function, and $\tau$ is the IoU threshold (*e.g.*, $\{0.53, 0.35, \ldots, 0.95\}$) for determining whether a prediction is considered a match to a ground truth. Similar to Mean IoU@$k$, Mean Recall@$k$ is also computed only on queries with at least $k$ ground-truth moments.

Taken together, these new metrics form a comprehensive and scalable evaluation framework that effectively measures the MMR task while maintaining strong compatibility with the SMR criteria. In particular, **G-mAP, mIoU@1, and mR@1** remain fully consistent with the standard SMR metrics, ensuring direct comparability of performance in both single- and multi-moment retrieval settings.

## 4 Methodology

As shown in Figure 3, the proposed FlashMMR model extends traditional SMR pipeline by introducing a novel Post-Verification Module. This module effectively adapts the existing framework for addressing MMR tasks. The FlashMMR consists of three key components: Feature Extraction and Fusion, Multi-Scale Temporal Processing, and the Post-Verification.

### 4.1 Feature Extraction and Multi-Scale Temporal Processing

Different with Single-Moment Retrieval, the goal is to locate a multiple video segment that best matches a textual query. Given a video $\mathcal{V}$ with clip-level features $\mathbb{V} = \{v_i\}_{i=1}^{L_v}$ and a query $\mathcal{Q}$ with word features $\mathbb{Q} = \{q_i\}_{i=1}^{L_q}$, the model predicts a set of temporal spans $(s_i, e_i, c_i)_{i=1}^n$, where $(s_i, e_i)$ defines the $i$-th predicted moment and $c_i$ is its confidence score.

**Feature Extraction.** Consistent with previous research [12, 20, 21], we extract video features $\mathbb{V}$ using frozen SlowFast [4] and CLIP [26] encoders, while text features $\mathbb{Q}$ are derived from CLIP.

The input video is segmented into clips at a predefined frame rate $r$ (*e.g.*, 0.5 FPS), and each clip is transformed into a feature representation $\{v_i \in \mathbb{R}^d\}_{i=1}^{L_v}$, while each query word is encoded as $\{q_i \in \mathbb{R}^d\}_{i=1}^{L_q}$. Both modalities are projected into a shared space of dimension $d$ via MLPs. $L_v$ and $L_q$ denote the video clip number and query word count, respectively. The dummy token and dummy encoder used here are identical to those in [2, 21], and act as explicit sinks that absorb semantics irrelevant to the query. Further implementation details are provided in the supplementary material.

**Cross-Feature Alignment.** To enhance video-text feature alignment, we adopt a Adaptive Cross Attention (ACA) module [21], which integrates learnable dummy token to encode contextual information beyond explicit query representations. After we get the fused feature $F \in \mathbb{R}^{L_v \times d}$ from ACA, it is further refined using a Transformer Encoder for long-range dependencies.

**Temporal Feature Layering.** To capture temporal variations across different moment durations, we employ a multi-scale temporal processing. We construct a temporal feature pyramid by applying a series of 1D convolutions to $F$:

$$F_p = \begin{cases} F, & \text{if } p = 1, \\ \text{Conv1D}^{p-1}(F, \text{stride} = 2), & \text{if } p = 2, 3, \ldots, P. \end{cases} \tag{1}$$

This results in a set of downsampled fused feature maps $\{F_p | p = 1, 2, \ldots, P\}$, capturing temporal dependencies at different granularities. Moment boundary predictions are computed at each scale using a shared convolutional head: $B_p = \sigma \left( \text{Conv1D} \left( \sigma \left( \text{Conv1D}(F_p) \right) \right)^\top \right)^\top \times C_p$, where $B_p \in \mathbb{R}^{\frac{L_v}{2^{p-1}} \times 2}$ represents boundary predictions, $C_p$ is a learnable scaling parameter, and $\sigma$ is the activation function.

**Adaptive Score Refinement.** To improve moment retrieval confidence, we refine moment scores using both intra-scale and inter-scale scoring: $c_p = \text{ScoreHead}_1(F_p)$, $p = 1, 2, \ldots, P$, $c_{\text{intra}} = \text{Concat}(c_1, c_2, \ldots, c_P)$, $c_{\text{inter}} = \text{ScoreHead}_2(\text{Concat}(F_1, F_2, \ldots, F_P))$. The final confidence score is computed as: $c_{\text{final}} = x \cdot c_{\text{intra}} + (1 - x) \cdot c_{\text{inter}}$, where $x$ is a learnable weighting factor.

We first obtain initial moment predictions. To better address the multi-moment setting, we introduce a post-verification module that enforces consistency across semantically related moments. This refinement leads to more diverse predictions and broader coverage in the multi-moment setting.

## 4.2   Post Verification Module

Post Verification Module refines initial predictions and improves alignment with the ground truth. It consists of two key components: Post-Processing with Feature Refinement and Semantic Consistency Control. We describe each component in detail below.

**Post-Processing with Feature Refinement.** Given the initial boundary predictions $\hat{B} \in \mathbb{R}^{3 \times n}$, where each predicted moment $\hat{b}_i = (s_i, e_i, c_i)$ consists of start time $s_i$, end time $e_i$, and confidence score $c_i$, we apply a structured post-processing strategy inspired by prior refinement techniques [2, 21]. This step ensures that the predicted windows adhere to temporal constraints and enhances their interpretability. We employ a post-processing function $\mathcal{F}(\cdot)$ parameterized by a set of structured constraints, including minimum and maximum window lengths, temporal clipping, and rounding heuristics:

$$\tilde{B} = \mathcal{F}(\hat{B}, \lambda_{\text{clip}}, \lambda_{\text{round}}), \tag{2}$$

where $\lambda_{\text{clip}}$ and $\lambda_{\text{round}}$ control boundary clipping and discretization granularity. This operation prevent excessively short or long predictions and align segments with predefined frame sampling rates. Following this step, each predicted interval is used to extract its corresponding multi-modal feature representation from the fused video embeddings $F \in \mathbb{R}^{L_v \times d}$: $\mathbf{I}_i = F[s_i \times r : e_i \times r, :]$, where the feature segments $\mathbf{I}_i$ are sampled based on the refined start and end timestamp.

**Post Verification via Semantic Consistency Control.** To re-evaluate the quality of predicted moments, we introduce a post-verification network $\mathcal{P}(\cdot)$, which models semantic consistency between retrieved intervals and their relevance to the query. This network is implemented as a recurrent module $\mathcal{P}_{\text{GRU}}(\cdot)$ [3], capturing contextual dependencies across extracted moment representations: $p_i = \sigma(\mathcal{P}_{\text{GRU}}(\mathbf{I}_i))$, where $p_i$ represents the refined confidence score assigned to each predicted moment, and $\sigma(\cdot)$ is the activation function. The refined score $\mathbf{p} \in \mathbb{R}^n$ provides an confidence adjustment that used to mitigate errors in the initial moment retrieval process.

Table 2: **Cross-Dataset Performance Comparison of SMR and MMR on QVHighlights and QV-M².** Experimental results for each method under three settings: (i) trained and evaluated on QVHighlights, (ii) trained on QVHighlights and evaluated on QV-M², and (iii) trained on QV-M² and evaluated on QVHighlights. The table reports the performance gains on both SMR and MMR tasks introduced by the new QV-M² dataset, and highlights the increased challenge that MMR poses to methods originally designed for SMR.

| Method | mAP | | | | mIoU@$k$ | | | mR@$k$ | | |
|---|---|---|---|---|---|---|---|---|---|---|
| | G-mAP | @1_tgt | @2_tgt | @3+tgt | @1 | @2 | @3 | @1 | @2 | @3 |
| M-DETR [12] | 32.79 | 42.02 | 19.45 | 3.67 | 48.81 | 32.75 | 28.54 | 40.19 | 24.56 | 19.55 |
| *w/ QV-M² Val* | 30.26 | 40.43 | 19.44 | 4.26 | 46.97 | 31.65 | 27.90 | 38.34 | 23.78 | 19.79 |
| *→ QV-M² Train* | **34.70** | **43.81** | **20.75** | **5.35** | **51.71** | **34.48** | **31.22** | **43.44** | **27.23** | **23.67** |
| EATR [9] | 35.96 | 44.15 | 23.80 | 7.70 | 50.91 | 36.44 | 34.00 | 42.85 | 30.14 | 27.65 |
| *w/ QV-M² Val* | 35.42 | 45.59 | 23.71 | 7.19 | 50.30 | 36.42 | 33.10 | 42.37 | 29.87 | 26.92 |
| *→ QV-M² Train* | **38.65** | **46.90** | **26.80** | **8.82** | **53.26** | **39.71** | **35.25** | **45.64** | **33.77** | **29.99** |
| UVCOM[35] | 42.83 | **51.71** | 29.96 | 11.02 | 57.79 | 40.92 | 38.79 | 51.01 | 35.26 | 32.97 |
| *w/ QV-M² Val* | 40.33 | 50.74 | 28.82 | 9.07 | 55.73 | 40.61 | 37.11 | 48.67 | 34.61 | 31.50 |
| *→ QV-M² Train* | **43.68** | **51.71** | **31.56** | **13.88** | **58.96** | **43.96** | **42.10** | **52.06** | **37.88** | **36.77** |
| QD-DETR [20] | 38.90 | 48.18 | 24.55 | 7.47 | 54.48 | 38.63 | **36.14** | 46.80 | 31.93 | **29.49** |
| *w/ QV-M² Val* | 36.32 | 46.62 | 24.82 | 6.97 | 52.75 | 37.53 | 33.81 | 45.01 | 30.70 | 27.21 |
| *→ QV-M² Train* | **40.63** | **49.94** | **26.91** | **8.69** | **56.39** | **39.76** | 36.09 | **48.72** | **32.96** | 29.42 |
| TR-DETR [31] | 36.86 | 46.20 | 24.63 | 5.18 | 53.86 | 36.59 | 31.18 | 45.33 | 28.91 | 24.13 |
| *w/ QV-M² Val* | 34.14 | 44.96 | 22.64 | 5.44 | 53.14 | 34.58 | 28.73 | 44.49 | 27.53 | 23.37 |
| *→ QV-M² Train* | **44.49** | **54.70** | **29.68** | **8.60** | **60.73** | **41.85** | **38.10** | **53.33** | **35.12** | **32.30** |
| FlashVTG [2] | 48.02 | 57.31 | 35.08 | 13.85 | 61.45 | 43.80 | 39.37 | 53.92 | 38.98 | 35.17 |
| *w/ QV-M² Val* | 40.28 | 50.21 | 29.93 | 9.37 | 57.4 | 44.03 | 39.42 | 49.92 | 37.24 | 32.91 |
| *→ QV-M² Train* | **48.35** | **57.37** | **35.40** | **15.71** | **62.61** | **44.86** | **41.83** | **55.67** | **40.03** | **37.77** |
| FlashMMR (Ours) | 48.07 | 56.95 | 35.78 | 15.15 | 62.09 | 45.32 | 40.32 | 55.02 | 40.63 | 36.68 |
| *w/ QV-M² Val* | 44.81 | 55.29 | 34.31 | 13.58 | 59.95 | 43.98 | 38.21 | 52.55 | 38.96 | 34.94 |
| *→ QV-M² Train* | **48.42** | **57.46** | **37.37** | **16.41** | **62.40** | **47.52** | **43.13** | **55.38** | **42.40** | **39.29** |

We supervise the refined score $\mathbf{p}$ by leveraging the temporal Intersection-over-Union (tIoU) between predicted segments and ground-truth moments. Given the ground truth annotations $B^* = \{(s_j^*, e_j^*)\}$, we compute the tIoU matrix and select the highest overlap score for each prediction:

$$\hat{\mathbf{IoU}} = \max(\text{tIoU}(\tilde{B}, B^*), \dim = -1).$$

By incorporating this post-verification module, our method effectively re-evaluates and refines initial moment predictions, leading to to more accurate and coherent grounding in the multi-moment setting.

### 4.3 Training Objectives

Follow the previous work [2], we employ Focal Loss [14], L1 Loss, and Clip-Aware Score Loss to optimize classification labels, temporal boundaries, and clip-level confidence scores, respectively. Additionally, we introduce a post-verification loss to refine moment predictions through a combination of mean squared error (MSE) loss and contrastive representation (Cross-Entropy) loss: $\mathcal{L}_{\text{PV}} = \|\mathbf{p} - \hat{\mathbf{IoU}}\|_2^2 + \mathcal{L}_{\text{repr}}$.

The representation loss $\mathcal{L}_{\text{repr}}$ enforces feature similarity consistency by encouraging temporally close frames to maintain high semantic coherence: $\mathcal{L}_{\text{repr}} = \sum_i \text{CE}(\mathbf{S}_i, \mathbf{T}_i)$, where $\mathbf{S}_i$ is a cosine similarity matrix computed over fusion features, and $\mathbf{T}_i$ represents the pairwise segment agreement derived from ground truth moment labels.

## 5 Experiments

### 5.1 Implementation Details

For fair comparison, we use SlowFast [4] and CLIP [26] as the video and text encoders, respectively, following configurations in [23]. FlashMMR and FlashVTG share identical parameter settings for common components. The post-verification loss terms $\mathcal{L}_{\text{PV}}$ and $\mathcal{L}_{\text{repr}}$ are weighted at 9 and 7. We use AdamW as the optimizer and set the NMS threshold to 0.7 during inference. SMR verification experiments on QVHighlights are conducted on the validation set due to the unavailability of test set annotations. All experiments are conducted on a single RTX 4090 GPU. Additional implementation details can be found in the supplementary material.

Table 3: **Comparison of performance on QV-M$^2$ test set with previous state-of-the-art methods.** The best results are highlighted in **bold**, and the second-best are underlined.

| Method | mAP | | | | mIoU@$k$ | | | mR@$k$ | | |
|---|---|---|---|---|---|---|---|---|---|---|
| | G-mAP | @1_tgt | @2_tgt | @3+tgt | @1 | @2 | @3 | @1 | @2 | @3 |
| M-DETR [12] *NeurIPS'21* | 20.65 | 33.71 | 25.85 | 10.95 | 44.14 | 38.98 | 34.34 | 34.81 | 30.95 | 26.24 |
| EATR [9] *ICCV'23* | 27.32 | 38.26 | 33.25 | 19.46 | 47.16 | 42.62 | 39.41 | 39.30 | 36.05 | 33.56 |
| QD-DETR [20] *CVPR'23* | 28.95 | 39.69 | 37.26 | 18.30 | 50.46 | 46.79 | 40.50 | 42.35 | 40.58 | 36.05 |
| TR-DETR [31] *AAAI'23* | 31.23 | 44.12 | 39.17 | 19.64 | 55.13 | 48.13 | 42.52 | 47.21 | 41.24 | 35.82 |
| CG-DETR [21] *Arxiv'24* | 28.87 | 43.74 | 32.35 | 18.44 | 52.01 | 47.98 | **43.27** | 43.26 | 40.80 | 36.69 |
| FlashVTG [2] *WACV'25* | 32.14 | 47.16 | 39.48 | 20.19 | 54.49 | 47.85 | 40.92 | 46.64 | 41.30 | 35.94 |
| **FlashMMR (Ours)** | **35.14** | **52.59** | **42.52** | **22.89** | **56.29** | **49.64** | 42.92 | **48.81** | **44.33** | **38.50** |

Table 4: **Comparison of performance on QVHighlights validation set with previous state-of-the-art methods.** The best results are highlighted in **bold**, and the second-best are underlined.

| Method | mAP | | | | mIoU@$k$ | | | mR@$k$ | | |
|---|---|---|---|---|---|---|---|---|---|---|
| | G-mAP | @1_tgt | @2_tgt | @3+tgt | @1 | @2 | @3 | @1 | @2 | @3 |
| M-DETR [12] *NeurIPS'21* | 32.79 | 42.02 | 19.45 | 3.67 | 48.81 | 32.75 | 28.54 | 40.19 | 24.56 | 19.55 |
| EATR [9] *ICCV'23* | 35.96 | 44.15 | 23.80 | 7.70 | 50.91 | 36.44 | 34.00 | 42.85 | 30.14 | 27.65 |
| QD-DETR [20] *CVPR'23* | 38.90 | 48.18 | 24.55 | 7.47 | 54.48 | 38.63 | 36.14 | 46.80 | 31.93 | 29.49 |
| TR-DETR [31] *AAAI'23* | 36.86 | 46.20 | 24.63 | 5.18 | 53.86 | 36.59 | 31.18 | 45.33 | 28.91 | 24.13 |
| CG-DETR [21] *Arxiv'24* | 43.69 | 52.70 | 30.12 | 10.02 | 60.32 | 45.04 | **42.21** | 52.85 | 38.17 | 35.46 |
| FlashVTG [2] *WACV'25* | 48.02 | **57.31** | 35.08 | 13.85 | 61.45 | 43.80 | 39.37 | 53.92 | 38.98 | 35.17 |
| FlashMMR (Ours) | **48.07** | 56.95 | **35.78** | **15.15** | **62.09** | **45.32** | 40.32 | **55.02** | **40.63** | **36.68** |

## 5.2 Comparison Results

We retrain and evaluate 6 methods on QV-M$^2$ and QVHighlights [12] dataset under both SMR and MMR settings. The experimental results, as presented in Tables 2, 3, 4, demonstrate the effectiveness of QV-M$^2$ for MMR and further show that FlashMMR consistently outperforms previous methods.

Table 2 presents Cross-Dataset Performance Comparison of SMR and MMR on QVHighlights and QV-M$^2$. Notably, models trained with QV-M$^2$ consistently exhibit improved performance compared to their counterparts trained only on QVHighlights, validating the effectiveness of QV-M$^2$ on both SMR and MMR supervision. FlashMMR achieves the highest overall G-mAP (48.42%) and superior performance across all mIoU@$k$ and mR@$k$ metrics. We also observe a performance drop across all methods when using QV-M$^2$ for evaluation, due to the increased number of one-to-many moment queries. This further demonstrates the effectiveness of our dataset in evaluating the MMR task.

To validate the effectiveness of our proposed FlashMMR, we compare it with state-of-the-art methods on the QV-M$^2$ and QVHighlights, as shown in Table 3, 4. FlashMMR achieves notable improvements over previous methods in most evaluation metrics, achieving a significant improvement over FlashVTG [2] in G-mAP (+3.00%), mAP@3+tgt (+2.70%), and mR@3 (+2.56%) on QV-M$^2$. Similar results can be observed in Table 4. These results highlight the superiority of our approach in localizing multiple relevant moments.

## 5.3 Ablation Study

We conduct an ablation study on the two MMR datasets—QV-M$^2$ and QVHighlights—to evaluate the effectiveness of the Post-Verification (PV) module in FlashMMR. As shown in Table 5, incorporating the PV module leads to consistent performance improvements across both datasets. On QV-M$^2$, the PV module brings a notable gain of 3.00% in G-mAP and 3.04% in mAP@2_tgt. Similar improvements are observed across other metrics, including mAP@3+tgt (+2.70%), mIoU@2 (+1.79%), and mR@2 (+3.03%), demonstrating its effectiveness in handling dense one-to-many queries.

On QVHighlights, which is comparatively less challenging, the PV module still yields consistent gains, including a 0.70% improvement in mAP@2_tgt and a 1.52% increase in mIoU@2. These results validate the robustness of the PV module and highlight its role in enhancing temporal consistency and filtering low-confidence predictions, ultimately improving multi-moment retrieval performance.

Table 5: Ablation study of the Post Verification (PV) module on MMR task.

| Dataset | FlashMMR | G-mAP | mAP@2_tgt | mAP@3+tgt | mIoU@2 | mIoU@3 | mR@2 | mR@3 |
|---|---|---|---|---|---|---|---|---|
| QV-M$^2$ | w/o PV | 32.14 | 39.48 | 20.19 | 47.85 | 40.92 | 41.30 | 35.94 |
| | w/ PV | **35.14** | **42.52** | **22.89** | **49.64** | **42.92** | **44.33** | **38.50** |
| QVHighlights | w/o PV | 48.02 | 35.08 | 13.85 | 43.80 | 39.37 | 38.98 | 35.17 |
| | w/ PV | **48.07** | **35.78** | **15.15** | **45.32** | **40.32** | **40.63** | **36.68** |

# 6 Discussion and Limitation

Experimental results show that while existing SMR models struggle to generalize to the MMR setting, FlashMMR improves overall performance and establishes a strong baseline, with QV-M$^2$ serving as a reliable testbed for advancing MMR research.

Despite these improvements, several challenges remain. First, our verification module remains in an early stage. Future work could explore more strategies, such as reinforcement learning or contrastive learning for better moment discrimination, to further enhance model performance. Secondly, one limitation in MMR research is the relatively limited size of high-quality annotated datasets. Although QV-M$^2$ is sufficient for current models, its limited scale may constrain future progress as models become more advanced.

# 7 Conclusion

In this paper, we revisited the gap between existing single-moment retrieval methodologies and the practical complexities inherent in real-world video understanding tasks. We introduce QV-M², the first fully human-annotated dataset for multi-moment retrieval, along with new evaluation metrics to benchmark the task. To address the limitations of traditional SMR frameworks, we proposed FlashMMR, a dedicated multi-moment retrieval model equipped with a Post-verification module. Comprehensive experiments demonstrate that FlashMMR effectively surpasses existing state-of-the-art moment retrieval methods, emphasizing the necessity and potential of future multi-moment retrieval frameworks. The proposed framework and dataset lay a groundwork for future research on more realistic and complex video temporal grounding tasks.

# Acknowledgment

This work is supported by Australian Research Council (ARC) Discovery Project DP230101753.

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
