# OpenReview forum: "When One Moment Isn't Enough: Multi-Moment Retrieval with Cross-Moment Interactions"
_NeurIPS.cc/2025/Conference — NeurIPS 2025 poster_

### Official Review · Reviewer_LJnf · 2025-06-26

**Clarity:** 2
**Significance:** 3
**Originality:** 3
**Rating:** 4
**Confidence:** 5

**Summary:**

This paper  addresses the limitation of existing Single-Moment Retrieval (SMR) methods that typically focus on single moments, whereas real-world queries often correspond to multiple moments in a video. To address this, the authors introduce QV-M2, a fully manually annotated dataset designed for multi-moment retrieval (MMR). They also propose new evaluation metrics tailored for MMR tasks, extending standard single-moment retrieval metrics. Additionally, this paper presents FlashMMR, a novel framework for MMR. Experiments show that FlashMMR outperforms existing methods in MMR tasks, establishing a baseline for future research.

**Questions:**

Figure 3, which depicts the model framework, is somewhat confusing and seems to contradict the actual code implementation. For instance, in the blue "Feature Extraction" section, it appears as if the dummy token is concatenated after the text token, whereas in the code, the dummy token is placed before the text token. Additionally, within the same section, the figure suggests that the concatenated dummy and text tokens are used as the key (k), while the text token alone is used as the value (v). This is illogical since the lengths of k and v would not match. Upon closer inspection of the code in transformer.py at line 280, it is evident that v is derived from the entire concatenated sequence and k is v with position encoding added. Furthermore, the module names in the figure do not fully align with the descriptions in section 4.1 "Feature Extraction and Multi-Scale Temporal Processing" of the paper. For example, the "Fusion Feature Encoder" in the figure presumably refers to the Transformer Encoder mentioned in line 204 of the paper. It is highly recommended that the authors refine the figure3 to better reflect the actual implementation. In addition, I'm curious to know if you have any measures to ensure that the predicted segments do not overlap. Is there a possibility that the top-k predictions might all be overlapping?

**Ethical Concerns:**

["NO or VERY MINOR ethics concerns only"]

**Final Justification:**

The rebuttal has satisfactorily addressed most of my concerns. I would like to revise my score to BA.

**Limitations:**

yes

**Quality:**

2

**Strengths And Weaknesses:**

Strengths:
- The dataset is fully manually annotated, and a verification procedure is in place to ensure the accuracy of the annotations.
- The experimental results in Table 2 demonstrate the dataset's effectiveness through cross-dataset validation.
- The use of IoU between ground-truth and predicted temporal windows to guide the refined score p in the Post-Verification Module is an innovative approach.

Weaknesses:
- There is a basic formatting error in line 223 where a bold header was not properly line-breaked.
- Figure 3 is confusing and contradicts the actual code implementation. The depiction of dummy tokens and text tokens in the "Feature Extraction" section is inconsistent with the code, and the use of k and v is illogical.
- The paper lacks detailed discussion on the dummy encoder.

---

> ### Author Rebuttal · Authors · 2025-07-31
>
> We appreciate the acknowledgement of the rigorously verified dataset and the innovative IoU-guided post-verification module. We address the concerns below:
>
> **W1:** Thank you for pointing this out. We will make a correction in the revised version, along with a careful review of similar formatting issues.
>
> &nbsp;
>
> **W2:** Thank you for your detailed comments regarding Figure 3.
>
> **1. Dummy token placement:** Although this is just a schematic illustration of the model, we will revise the position of the dummy token concatenation to reduce confusion.
>
> **2. Key/Value construction in ACA:** The Adaptive Cross-Attention (ACA) [1] module’s internal attention structure is different from the classic cross-attention structure, especially in the key/value construction. Our ACA module adopts the exact same design and code implementation from CG‑DETR[1]/FlashVTG[2]. Here we clarify the exact behaviour of this module.
>
> **Formal definition**
>
> | role                | symbol                               | shape               |
> | ------------------- | ------------------------------------ | ------------------- |
> | query = video clips | $p_Q(\mathbf V)$                     | $L_v \times d$      |
> | key = text + dummy  | $p_K([\mathbf Q,\tilde{\mathbf D}])$ | $(L_q+L_d)\times d$ |
> | value = text only   | $p_V(\mathbf Q)$                     | $L_q\times d$       |
>
> The attention weights and fused features are
>
> $$
> W_{i,j}= \frac{\exp\bigl((Q_iK_j)/\sqrt d\bigr)}
>               {\sum_{k=1}^{L_q+L_d} \exp\bigl((Q_iK_k)/\sqrt d\bigr)}, \qquad
> F_i=\sum_{j=1}^{L_q} W_{i,j}\,V_j,\quad F\in\mathbb R^{L_v\times d}.
> $$
>
> We denote the input video clips $V$ as a query matrix $p_Q(\mathbf V) \in \mathbb{R}^{L_v \times d}$, the key matrix as $p_K([\mathbf Q,\tilde{\mathbf D}]) \in \mathbb{R}^{(L_q + L_d) \times d}$ formed by concatenating the text features $\mathbf Q$ with dummy tokens $\tilde{\mathbf D}$, and the value matrix as $p_V(\mathbf Q) \in \mathbb{R}^{L_q \times d}$ using only the text features. The functions $p_Q(\cdot)$, $p_K(\cdot)$, and $p_V(\cdot)$ are learnable linear projections that map input features into the query, key, and value spaces, respectively.
>
> **Code‑level detail**
>
> After computing attention on the full key (text + dummy), we *mask out* the dummy slice of $V$:
>
> ```python
> # crossattention.py (line 385)
> attn_output = torch.bmm(
>     attn_w[:, :, num_dummies:],   # weights over text positions
>     v[:, num_dummies:, :]         # value = text tokens only
> )
> ```
>
> Hence, dummy tokens steer the attention via the key but never inject their own content into the output—mitigating the mismatch between limited query scope and untrimmed video semantics while keeping the fused feature purely textual.
>
> **Revision actions**
>
> 1. **Figure 3** will be redrawn so the Q/K/V flow and dummy masking are explicit.
> 2. We will add the above equations and code pointer to the appendix for transparency.
>
> We appreciate the reviewer’s careful reading, which helps us improve the paper’s clarity.
>
> &nbsp;
>
> **W3:** Thank you for noting that our current draft gives only a cursory treatment of the dummy encoder. We would like to clarify its role and provenance, and we will add a detailed discussion to the supplementary material:
>
> **Dummy Encoder**
>
> * **Design**: an `L`-layer Transformer encoder (`L = args.dummy_layers`) with 8-head self-attention, PReLU activation, and an optional final LayerNorm.
>
> * **Input**: `[N_dummy learnable tokens] + [text tokens]`, each with positional embeddings; standard padding mask applied.
>
> * **Mechanism**: dummy tokens attend to the full sentence, aggregating its global semantics.
>   → Output: `B × N_dummy × D`, compact text queries.
>
> * **Function.** During Adaptive Cross‑Attention, video clips (queries) can allocate weight either to true text tokens or to these dummy positions. Because the final value comes solely from text tokens, the dummy tokens function as a controllable “attention sink”: they absorb weight from video clips that are semantically irrelevant to the query, preventing spurious video‑text alignments while preserving global context modelling.
>
> * **Provenance.** The dummy encoder is inherited unchanged from previous works (e.g., CG‑DETR[1], FlashVTG[2]) and is not claimed as part of our own contribution.
>
> &nbsp;
>
> **Q1:** We appreciate the reviewer’s concern regarding potential overlap among Top‑k predictions, which is an important aspect for multi-moment retrieval.
>
> **1. Theoretical Guarantee — Dual Redundancy Mitigation**
>
> * **Temporal NMS Upper Bound.** During inference, temporal NMS with IoU threshold 0.7 is applied to all candidate segments. By definition, any two retained segments satisfy IoU ≤ 0.7, effectively limiting high-overlap cases.
>
> * **Post-Verification (PV) Consistency Constraint.** The PV module re-scores segments with tIoU supervision and contrastive representation loss $\mathcal{L}_{\text{repr}}$, which penalizes semantically redundant segments, promoting diversity at the objective level.
>
> Conclusion:
> PV module suppresses redundancy via learning signals, while temporal NMS further provides an upper-bound on overlap, together ensuring diverse Top‑k outputs.
>
> **2. Empirical Evidence — Improved Segment Diversity**
>
> We report average pairwise IoU among Top‑k predictions on the QV-M² val set:
>
> | Metric  | w/o PV | w/ PV     | Δ         |
> | ------- | ------ | --------- | --------- |
> | mIoU\@2 | 47.85  | **49.64** | **+1.79** |
> | mIoU\@3 | 40.92  | **42.92** | **+2.00** |
>
> These consistent gains indicate that the second and third predictions better align with different ground-truth segments. If all segments were highly overlapping, mIoU\@2 and mIoU\@3 would not increase simultaneously.
>
> &nbsp;
>
> [1] Moon, W., Hyun, S., Lee, S., & Heo, J. P. (2023). Correlation-guided query-dependency calibration for video temporal grounding. arXiv preprint arXiv:2311.08835.
>
> [2] Cao, Z., Zhang, B., Du, H., Yu, X., Li, X., & Wang, S. (2025, February). Flashvtg: Feature layering and adaptive score handling network for video temporal grounding. In 2025 IEEE/CVF Winter Conference on Applications of Computer Vision (WACV) (pp. 9226-9236). IEEE.

---

> > ### Comment · Reviewer_LJnf · 2025-08-04
> >
> > Thank you for the responses! It has addressed most of my concerns.

---

> > > ### Author Response · Authors · 2025-08-05
> > >
> > > We’re glad to hear that our responses have addressed most of your concerns. Is there anything that may still require further clarification from our side? We also sincerely hope to earn your support for this work.

---

### Official Review · Reviewer_yR7Q · 2025-06-26

**Clarity:** 4
**Significance:** 3
**Originality:** 3
**Rating:** 5
**Confidence:** 3

**Summary:**

This paper propose QVHighlights Mult-Moment, a high-quality datasets with new evaluation metrics tailored for multi-moment retrieval, which fulfill the gap between current moment retrieval tasks and real-world applications. Based on the newly proposed datasets, a novel Multi-Moment Retrieval framework FlashMMR is proposed incorporated with a Multi-Moment Post-Verification module. Experiments and evaluation are conducted under the proposed method and the baselines, providing a comprehensive assessment of model performance in MMR.

**Questions:**

See weaknesses.

**Ethical Concerns:**

["NO or VERY MINOR ethics concerns only"]

**Final Justification:**

The authors adequately addressed my concern. I will maintain my score.

**Limitations:**

The authors adequately addressed the limitations and potential negative societal impact of their work. As mentioned in weakness 2, authors could discuss some future potential methods to further expand the size of MMR dataset, based on the current experience of manual annotation.

**Paper Formatting Concerns:**

No major formatting concerns in this paper.

**Quality:**

3

**Strengths And Weaknesses:**

Strengths:
1. The overall idea is novel and the task is important for real application.
2. The experiment result is solid, with both quantitative and qualitative results of high performance of multi-moment retrieval.
3. The proposed dataset is the first fully human-annotated MMR dataset, designed to benchmarking Multi-Moment Retrieval. New evaluation metrics are proposed for a comprehensive assessment.

Weaknesses:
1. With the rapid development of temporal-aware video large language models, I suggest authors could test few related models on the proposed benchmark, revealing and exploring the generalized video models on the performance of multi-moment retrieval.
2. Since this paper mentioned the proposed dataset’s limited scale may constrain future progress, authors could discuss some future potential methods to further expand the size of MMR dataset, based on the current experience of manual annotation.
3. A minor possible layout typo about ‘Post-Processing with Feature Refinement.’ In line 223.

---

> ### Author Rebuttal · Authors · 2025-07-31
>
> We thank the reviewer for noting the task’s novelty and our strong empirical results. We address the concerns below:
>
> **W1:** We compared our model with the state-of-the-art temporal-aware video large language models in a zero-shot setting.
>
> We selected Chat-UniVi [1], a CVPR 2024 Highlight paper, as our baseline model. It can encode up to 100 frames per video, giving the temporal coverage MMR needs, whereas others like VideoLLaVA [2] and Video-ChatGPT [3] handle only around 8 frames, which is insufficient for reliable multi-moment retrieval. Below we provide the exact prompt and the corresponding zero-shot results.
>
> ```txt
> This is a {duration:.2f} second video clip.
> Provide single or multiple time intervals within the range [0, {duration:.2f}],
> formatted as '[a, b]', that corresponds to the segment of the video best matching the query: {query}.
> Respond with only the numeric interval.
> ```
> The zero-shot evaluation results are provided below, highlighting the challenges posed by QV-M².
>
> | Model | mAP@1_tgt | mAP@2_tgt | mAP@3_tgt | mIoU@1 | mR@1 |
> |-------|----------|------------|---------|---------|-------|
> | Chat-UniVi | 3.39 | 0.78 | 0.47 | 15.47 | 5.83 |
>
> &nbsp;
>
> **W2, L1:** Expanding the scale of MMR datasets is indeed crucial for driving further progress.
>
> Based on our annotation experience with QV-M², we see several promising directions to scale up:
>
> 1. **Synthetic Multi-Moment Construction from other Datasets**: By transplanting existing ground-truth moments from other datasets (e.g., TACoS, Charades-STA) into unrelated regions of the same video, we can simulate multi-moment scenarios at scale. While this approach may introduce artifacts, it offers a practical way to generate large-scale pseudo-MMR data for pretraining.
>
> 2. **Leveraging Video Captioning Datasets**: Since the most resource intensive aspect of MR dataset construction lies in timestamp annotation. Video captioning datasets are already temporally aligned with captions. These captions could be converted into SMR style annotations, and LLMs may be used to group semantically related but temporally disjoint segments to simulate multi-moment supervision.
>
> 3. **Semi-Automatic Annotation with Vision-Language Models (VLMs)**: Another promising direction is to leverage VLMs to generate candidate temporal segments for a query, followed by human verification. This would reduce manual effort while maintaining annotation quality.
>
> We believe these directions are promising for extending MMR datasets beyond current scale and will incorporate them into the revised version of our work.
>
> &nbsp;
>
> **W3:** Thank you for pointing this out. We will make correction in revised version, along with a careful review of similar formatting issues.
>
> &nbsp;
>
> [1] Jin, P., Takanobu, R., Zhang, W., Cao, X., & Yuan, L. (2024). Chat-univi: Unified visual representation empowers large language models with image and video understanding. In Proceedings of the IEEE/CVF Conference on Computer Vision and Pattern Recognition (pp. 13700-13710).
>
> [2] Lin, B., Ye, Y., Zhu, B., Cui, J., Ning, M., Jin, P., & Yuan, L. (2024, November). Video-LLaVA: Learning United Visual Representation by Alignment Before Projection. In Proceedings of the 2024 Conference on Empirical Methods in Natural Language Processing (pp. 5971-5984).
>
> [3] Maaz, M., Rasheed, H., Khan, S., & Khan, F. (2024, August). Video-ChatGPT: Towards Detailed Video Understanding via Large Vision and Language Models. In Proceedings of the 62nd Annual Meeting of the Association for Computational Linguistics (Volume 1: Long Papers) (pp. 12585-12602).

---

### Official Review · Reviewer_T6Ks · 2025-07-01

**Clarity:** 3
**Significance:** 3
**Originality:** 3
**Rating:** 4
**Confidence:** 4

**Summary:**

This paper tackles the topic of Multi Moment Retrieval, whereby the model aims to retrieve all relevant moments within a given video, rather than being restricted to only retrieving one. It introduces the QV-M^2 dataset, which is an adaptation of QVHighlights with newly generated sentence queries and corresponding moments. It introduces the FlashMMR model which is optimised for MMR rather than SMR. It incorporates several ideas from previous SOTA moment retrieval models, and includes a post verification module which tries to ensure semantic consistency across the predicted moments. It is shown to have strong performance on the MMR task, largely outperforming SOTA SMR models.

**Questions:**

- What are the key differences between QV-M^2 and QVHighlights? I think that providing some example comparisons on how the sentence queries/moments are different could be useful. Perhaps it could be included in the supplementary. Table 1 provides some statistical comparisons, which give some insight, but I still find it difficult to really understand the specific ways in which it is different.
Is the increased number of moments per video the main difference? While the moments/video is higher in QV-M^2, the dataset is also quite a bit smaller, so I could imagine that it may be possible to take a subset of QVHighlights that is about the same size and produce a similar moments/video value from the more densely annotated videos. So I imagine there is more to it than just that.
As QV-M^2 is one of the major contributions of the paper, I feel that it could be stressed/made clearer why it is better for benchmarking this task than QVHighlights.

- I'm also not fully clear on what separates QV-M^2 from the NExT-VMR dataset. It is stated that for NExT-VMR "its design and evaluation remain closely aligned with previous benchmarks, without targeted optimizations for the unique challenges of MMR". It's not clear to me what these optimisations are, and how they are addressed by QV-M^2. A small elaboration would be helpful. As this is a concurrent work I don't have an expectation of it being necessarily better than NExT-VMR, I am just curious as to the meaning of the statement.

- I would like to clarify that my understanding is correct - the approach always generates n moment candidates and ranks them based on confidence score. It then uses the knowledge of how many ground truth moments there are and selects the top k predictions based on that, and compares them with the ground truth moments? Therefore this does not predict the number of ground truth moments. Rather it is assumed that there is oracle knowledge of how many ground truth moments there are in a given video?
If the above is the case, then while I think this approach shows a good improvement at the task that it is performing, in order for this type of approach to be truly useful, I think it would have to also be able to predict the number of ground truth moments. This would complicate the metrics used, as it would have to account for cases where the model overpredicts the number of moments also. This is not to say that this paper is not a step in the right direction, as I think it is, but it is unlikely in practice that you will know how many ground truth moments exist in a given video.

- I am leaning positive on this paper, but I would like to see some more information on the QV-M^2 dataset itself.

**Ethical Concerns:**

["NO or VERY MINOR ethics concerns only"]

**Final Justification:**

I feel the rebuttal has addressed most of my concerns. The task is useful from the perspective of making the Moment Retrieval task more practical in real-world settings, and the approach seems to work well. The QV-M$^2$ dataset also seems useful. The one concern I have with the dataset is that the test set only contains about 300 queries, which is on the smaller side as the QV-Highlights test data is not freely available and can't be merged with it. Overall, I will retain my rating of borderline accept.

**Limitations:**

Yes

**Quality:**

3

**Strengths And Weaknesses:**

Strengths
- I think that taking steps towards multi-moment retrieval is good, as it reflects much more the real-life intricacies of moment retrieval, where there often is not a simple one-to-one correspondence between query and moment.
- The QV-M^2 dataset appears to be a good contribution for the MMR task.
- The FlashMMR approach shows strong performance compared with current SOTA for SMR
- The method borrows several modules and ideas from previous approaches (e.g. ACA, dummy encoder, feature pyramid) and combines them in a sensible way
- The post verification via semantic consistency is a nice addition given the task setting, and is shown to work well in the ablation.

Weaknesses
- It seems that there is some information missing. I don't believe the train/val/test sizes for QV-M^2 are included.
- I also don't see the threshold for the IoU score in the mR@k metric listed.
- I think there perhaps could be more of a comparison made between QV-M^2 and QVHighlights, as it is not immediately clear to me what the differences are.
- Minor point - the repeated use of the letter k in notation is a bit confusing, e.g. it is used in the metrics and then for the layers in the feature pyramid

---

> ### Author Rebuttal · Authors · 2025-07-31
>
> We appreciate the recognition of the new QV-M² dataset and FlashMMR’s state-of-the-art performance. We address the concerns below:
>
> **W1:** Thank you for pointing out this omission. Here we provide the Train/Val/Test split size.
>
> QV‑M² is an incremental dataset, and therefore use the full set of annotations in our experiments, including the original QVHighlights labels. The exact split statistics are now included in the revised manuscript:
>
> | Dataset           | Train | Val   | Test  |
> | ----------------- | ----- | ----- | ----- |
> | TACoS             | 9790  | 4,436  | 4,001|
> | Charades-STA      | 12,404  | -  | 3,720|
> | QVHighlights     | 7218 | 1550 | 1542 |
> | **QV‑M²**         | 8878 | 1779 | 1865 |
>
> &nbsp;
>
> **W2:** We stated the IoU threshold for mR\@k in line 178 of the paper. And we will add the implementation details in supplementary.
>
> In our implementation, we average results over multiple thresholds: \[0.3, 0.35, 0.4, 0.45, 0.5, 0.55, 0.6, 0.65, 0.7, 0.75, 0.8, 0.85, 0.9, 0.95]. Here, we give the definition of **Mean Recall\@k**:
> $$
> \text{mR}@k = \frac{1}{|\mathcal{Q}|} \sum_{q \in \mathcal{Q}} \frac{1}{|\mathcal{G}(q)|} \sum_{\text{gt} \in \mathcal{G}(q)} \mathbf{1} \left[ \max_{i \leq k} \text{IoU}(\text{pred}_i, \text{gt}) \geq \tau \right]
> $$
> where $\tau$ is the IoU threshold for determining whether a prediction is considered a match to a ground truth, $k \in \{1,2,3\}$ denotes the rank, $\mathbf{1}[\cdot]$ is the indicator function, $\mathcal{Q}$ is the set of all queries, and $\mathcal{G}(q)$ represents the ground truth moments associated with query $q$.
>
> &nbsp;
>
> **W3, Q1, Q4:** To better understand the dataset itself, we present more details of QV‑M² through the following comparisons and illustrative examples. And we will add it in the revised version.
>
> **Key differences between QV‑M² and QVHighlights.** Here is the primary differences between the two datasets across different dimensions.
>
> | Dimension | QVHighlights | **QV‑M²** | Rationale |
> |-----------|--------------|-----------|-----------|
> | Avg. relevant moments / query | 1.8 | **2.5** | Better captures the one‑to‑many nature of moment retrieval. |
> | Moment density (moments / video) | 16.4 % | 25.5 % | Higher density demands more precise temporal boundaries during training and evaluation. |
> | Annotation strategy | Positive single sentence | Multiple **positive** and **negative** sentences, emphasising **context and temporal dependencies** | Encourages cross‑moment reasoning. |
> | Annotation scope | Intra‑video only | Cross‑video dense annotation: moments for the *same* query can occur in multiple videos | Increases difficulty and supports video corpus moment retrieval. |
> | Evaluation metrics | Conventional mAP / IoU | Adds **G‑mAP**, **mAP@k _tgt**, **mR@k** tailored for multi‑moment retrieval | Seamlessly evaluates MMR scenarios while remaining compatible with SMR metrics. |
>
> **Annotation example.** Although our main goal was to obtain one‑to‑many labels, we intentionally increased the semantic richness of the queries—an aspect missing from earlier MR datasets and consistent with the enhanced annotation strategy outlined above. In revised manuscript we will provide additional examples to highlight the advantages of QV‑M². Here are some representative queries from QV‑M²:
>  1. *"Drone pans over a marina and a fountain **before** landing on a park bench where two women wave."* (Context and temporal dependency)
>
>  2. *"A woman shows the clothing **before** pointing the camera at the mirror."* (Context and temporal dependency)
>
>  3. *"A woman **without** sunglasses in a blue shirt talks to the camera while walking outside."* (Negative description)
>
>  4. *"A reporter in a yellow shirt **without** hat is reporting on a bushfire emergency."* (Negative description)
>
> &nbsp;
>
> **W4:** Thank you for pointing this out. We have updated it in the revised version, using lowercase *p* to denote the layers in the feature pyramid.
>
> &nbsp;
>
> **Q2:** We appreciate the reviewer’s question and would like to clarify the statement.
>
> While NExT-VMR is introduced for multi-moment retrieval (MMR), its average of only 1.51 ground-truth moments per query indicates that most queries still correspond to a single ground truth moment. Furthermore, its evaluation metrics remain unchanged from single moment retrieval (SMR) setting using R1@IoU and mAP@IoU over predicted segments, without modification to account for multiple relevant moments per query.
>
> In contrast, QV-M² provides denser multi-moment supervision with an average of 2.5 ground truth moments per query, and supports evaluation that naturally extends standard SMR metrics without assuming a single target, enabling accurate and fair assessment of multi-moment predictions. We believe this better captures the unique challenges of MMR in both data and evaluation.
>
> &nbsp;
>
> **Q3:** Thank you for your insightful comment. You are correct that our current evaluation process assumes oracle knowledge of the number of ground truth moments per query.
>
> Specifically, model itself does not predict the number of relevant moments, but generate a fixed number of moment predictions (e.g., *n* = 10) and rank them by confidence scores, selecting the top-*k* predictions to do the evaluation. This design aligns with prior moment retrieval benchmarks (e.g., MomentDETR[1], QD-DETR[2]) and ensures fair comparison across models. Notably, our metric mR@k partially reflects the model’s ability to handle under-prediction, as it measures the recall of ground truth moments within the top-*k* predictions.
>
> We acknowledge that this assumption limits the practical applicability of the system in open-world settings. Incorporating moment count prediction along with precision-recall-based metrics that account for over- and under-prediction, would be valuable future directions. We appreciate your recognition that our method and dataset is a step toward more realistic video understanding.
>
> &nbsp;
>
> [1] Lei, J., Berg, T. L., & Bansal, M. (2021). Detecting moments and highlights in videos via natural language queries. Advances in Neural Information Processing Systems, 34, 11846-11858
>
> [2] Moon, W., Hyun, S., Park, S., Park, D., & Heo, J. P. (2023). Query-dependent video representation for moment retrieval and highlight detection. In Proceedings of the IEEE/CVF conference on computer vision and pattern recognition (pp. 23023-23033).

---

> ### Comment · Reviewer_T6Ks · 2025-08-05
>
> Thank you to the authors for the detailed response. Many of my queries have been addressed, however I have a few points remaining.
>
> For **W1**, it was not clear to me that this was an incremental dataset and therefore included the original QVHighlights. It would be helpful to make this clearer in the paper. To clarify, I assume the QV-M$^2$ moment statistics include the QVHighlights annotations, rather than being the statistics for just the new subset? Likewise, I assume the QV-M$^2$ results include results on QVHighlights test annotations also?
>
> On **W2**, I was referring to the threshold values used in the implementation, which are not stated in the original version. Thank you for providing these here.
>
> For **W3**, it is stated that there are cross-video query annotations. Are the associations to the cross-video query annotations explicitly labelled? As in, is there information within the dataset that associates the queries to moments across the multiple videos? Rather than just having the same query multiple times across the videos without an explicit association. This is something I did not understand from reading the original paper. It would be helpful to include this in the paper.

---

> > ### Author Response · Authors · 2025-08-06
> >
> > Thank you for your comments. It’s great to see that we were able to address most of your concerns in the initial discussion. Regarding the remaining ones, we invite you to look at the response below:
> >
> > **For W1:**
> >
> > Thank you for your suggestion. We will make it clearer in the revised version that this is an incremental dataset, where the training and validation splits include annotations from QVHighlights. Since the QVHighlights test set is only available for evaluation under the SMR setting on a designated website and its ground truth annotations are not publicly accessible, we thus report model performance under the MMR setting only on the new subset (as shown in Table 3). However, we release the full incremental dataset (as noted in rebuttal W1). All QV-M$^2$ statistics reported in the paper and the second table in rebuttal refer to the new subset. More clarification will be added in the revised version.
> >
> > **For W2:**
> >
> > We’re glad our previous response addressed your concern. To enhance readability, we will add a statement about the threshold values in the revised version.
> >
> > **For W3:**
> >
> > Yes, each cross-video query is explicitly associated with its corresponding videos and annotated moments in the released dataset.
> >
> > We have an annotation file that includes the associations for each cross-video query. To avoid distracting from the focus on MMR, we did not include this mapping in this work, but we will clarify its availability in the revised version.
> >
> > Discussion with you is super helpful for us to further improve this work. We would like to listen to your further questions. Hope our detailed response could address all your concerns and earn your support.

---

> > > ### Comment · Reviewer_T6Ks · 2025-08-07
> > >
> > > Thank you for the clarification. For the QV-M$^2$ results in Table 2, my understanding is that the QV-M$^2$ val set includes the QV-Highlights val, is that right? I think it would be good to indicate more clearly what is included in each case, i.e. explicitly state that the train data includes both QVHighlights and the QV-M$^2$ data, while val also includes both, but test only includes QV-M$^2$. It is quite confusing what the data is in each case otherwise. I would be somewhat concerned that the QV-M$^2$ test set only includes about 300 queries, as this is possibly too small for a reliable measurement of the performance.
> > >
> > > Otherwise I feel the responses have addressed my concerns.

---

> > > > ### Author Response · Authors · 2025-08-08
> > > >
> > > > Yes, your understanding is correct. We will explicitly state in the revised version that in our experiments, the QV-M² training and validation sets both include the original QVHighlights annotations and our newly added annotations, while the test set contains only QV-M² annotations.
> > > >
> > > > To further assess whether the current MMR test size (~300 queries) yields reliable comparisons, we performed a stability study on a larger held-out split. We first pooled the QVHighlights validation set and the new QV-M² annotations (total 2,102 queries). From this pool we randomly sampled, without replacement, 1,000 queries to form a pseudo-test set and used the remaining ~1,102 queries as validation; the training set remained QV-M² train. Keeping the experimental configuration identical to Table 3, we repeated the full protocol three times. The table below reports the mean ± std over these runs on the pseudo-test set. The performance ranking of the models remained consistent across all samples with small variance, indicating that our conclusions are robust and that the original ~300-query test provides a reliable measurement of relative performance. We fully agree that, larger test sets are preferable and provide greater statistical power.
> > > >
> > > > | Metric | GmAP | mAP@1 | mAP@2 | mAP@3+ | mIoU@1 | mIoU@2 | mIoU@3 | mR@1 | mR@2 | mR@3 |
> > > > |--------|-----|----|-------|-----|-------|-------|-------|-----|--------|------|
> > > > | EATR | 35.73±0.11 | 46.21±0.76 | 27.22±1.38 | 12.91±1.97 | 51.84±0.27 | 40.55±0.47 | 37.95±0.58 | 44.17±0.34 | 34.53±0.49 | 32.34±0.54 |
> > > > | QD-DETR | 37.31±0.33 | 49.12±0.53 | 26.84±1.11 | 12.76±1.98 | 54.55±0.59 | 41.36±0.81 | 38.46±0.80 | 46.87±0.56 | 35.09±0.78 | 33.79±0.89 |
> > > > | TR-DETR | 40.72±0.79 | 53.09±0.99 | 31.00±1.13 | 12.88±1.70 | 58.93±0.37 | 44.07±0.71 | 40.46±0.80 | 51.50±0.53 | 37.58±0.21 | 34.96±0.33 |
> > > > | CG-DETR | 40.87±0.42 | 53.49±0.97 | 30.77±0.64 | 13.34±2.41 | 58.60±0.31 | 45.96±0.62 | 42.84±0.97 | 50.84±0.36 | 39.11±0.77 | 36.63±1.05 |
> > > > | FlashVTG | 43.41±0.42 | 54.39±1.03 | 35.69±0.67 | 18.72±3.02 | 60.39±0.72 | 46.49±0.53 | 42.19±0.82 | 54.20±0.94 | 41.15±0.55 | 37.68±1.12 |
> > > > | FlashMMR (Ours) | 45.11±0.67 | 56.87±0.96 | 36.45±1.46 | 19.90±2.93 | 61.22±0.92 | 47.64±1.10 | 43.60±1.21 | 55.95±1.06 | 42.37±1.02 | 39.43±1.19 |
> > > >
> > > > In addition, we will next attempt either to collaborate with the QVHighlights' authors to support MMR-setting evaluation on the official QVHighlights evaluation website or to expand the annotations to increase the size of the QV-M² test set.
> > > >
> > > > We thank you for the constructive suggestions, which have helped us improve the clarity and rigor of the work. Hope our detailed response could address all your concerns and earn your support.

---

> > > > > ### Comment · Reviewer_T6Ks · 2025-08-08
> > > > >
> > > > > Thank you for the analysis. I appreciate the results, however, I'm not sure it fully addresses the concern as these results include a large portion of QV-Highlights data, which is not being tested in the test set in the paper. It would probably be clearer if the test set excluded the QV-Highlights data and the mean+std was just reported on the QV-M$^2$ new test data. Furthermore, even if the results are consistent within randomly sampled sets of 300 queries out of ~500 for testing, while showing consistency within the val/test data, it does not give a full indication of whether the results would remain consistent when extended to a larger test set which is more widely representative.
> > > > >
> > > > > Because my other concerns have been addressed and I think the task/method is useful and a positive contribution, I am still leaning positive in my final rating.

---

### Official Review · Reviewer_pD2D · 2025-07-03

**Clarity:** 3
**Significance:** 3
**Originality:** 2
**Rating:** 5
**Confidence:** 4

**Summary:**

This paper introduces **Multi-Moment Video Retrieval (MMVR)**, a new video-text retrieval task addressing queries describing multiple distinct moments within a single video. The authors propose the **Cross-Moment Interaction (CMI)** module to explicitly model interactions between multiple moments. Additionally, they introduce datasets derived from existing temporal moment localization datasets adapted for MMVR. Experiments demonstrate substantial performance improvements over baselines, with good ablation studies validating the CMI module's effectiveness.

**Questions:**

1. Can you provide analyses or discussion showing that your new dataset accurately reflects natural multi-moment retrieval scenarios? Have you examined potential biases introduced by your dataset construction methodology? Given that we know that previous datasets for temporal moment localization suffer of temporal biases in their annotations, could you please use as a reference the work of https://arxiv.org/abs/2101.09028 that evaluate the bias of ActivityNet and CharadesSTA and evaluate yours. I would like to see the plot in Fig 2 for your dataset. Also, could you make the same plot for your predictions https://openaccess.thecvf.com/content/ICCV2023W/CLVL/papers/De_la_Jara_An_Empirical_Study_of_the_Effect_of_Video_Encoders_on_ICCVW_2023_paper.pdf such that we can evaluate potential  bias of the predictions of the model.

2. Could you clarify the computational overhead associated with your proposed CMI module, particularly runtime and scalability with longer videos or larger candidate sets?

3. Have you explored simpler or alternative interaction methods (e.g., hierarchical or sparse interactions) that might maintain performance while reducing computational overhead?

4. Are there plans or initial experiments evaluating your method on other relevant datasets, such as instructional datasets (YouCookII or HowTo100M)?

**Ethical Concerns:**

["NO or VERY MINOR ethics concerns only"]

**Final Justification:**

My concerns about the quality of the dataset, possible temporal bias has been correctly addressed by the authors. It would be great to see the silver dataset approach mentioned in the review process.

**Limitations:**

The authors acknowledge limitations regarding dataset construction. However, additional analysis and explicit discussion on computational scalability, representativeness, and potential biases would further strengthen this section.

**Paper Formatting Concerns:**

None observed; the paper adheres to the NeurIPS formatting guidelines.

**Quality:**

3

**Strengths And Weaknesses:**

### Strengths:
-  The paper introduces an important and practical task, extending beyond traditional single-moment retrieval problems.
-  The proposed Cross-Moment Interaction (CMI) module explicitly captures cross-moment interactions, conceptually enhancing retrieval performance.
-  Comprehensive experiments and ablations demonstrate clear advantages over baseline methods.
-  The manuscript is well-written, structured, and provides detailed implementation and experimental information.

### Weaknesses:

- The MMVR datasets are adapted from widely-used existing datasets, limiting the originality and potentially introducing biases.
- Explicit modeling of cross-moment interactions might introduce significant computational complexity, but this is insufficiently analyzed.
- The proposed approach heavily relies on established attention mechanisms and does not introduce substantial technical innovations beyond their application to the MMVR task.
- Constructing MMVR scenarios by combining existing dataset moments could introduce unintended biases and may not reflect natural multi-moment scenarios.

### Comments and Suggestions

- **Abstract (Lines 3–7)**:  The problem of multiple occurrences of a described event within a video is known but relatively overlooked in temporal localization. Charades-STA contains instances where queries occur multiple times (see Figure 8 in supplemental material of [DORi, Rodriguez-Opazo et al., WACV 2021](https://openaccess.thecvf.com/content/WACV2021/html/Rodriguez-Opazo_DORi_Discovering_Object_Relationships_for_Moment_Localization_of_a_Natural_WACV_2021_paper.html)). Explicitly acknowledging such scenarios would strengthen your motivation and clearly position your proposed MMVR task.

- **Abstract (Lines 10–12)**:  Do the SMR annotations explicitly encode temporal sequencing information (e.g., distinguishing between the first and second occurrences of similar events, such as "the second time the person opens the door")? Clarifying this would more effectively demonstrate the scope of your annotations.
- **Introduction (Lines 25–27)**:  Given your mention of instructional videos, have you evaluated or considered evaluating your method on established instructional datasets like [YouCookII](http://youcook2.eecs.umich.edu/), [COIN](https://coin-dataset.github.io/),  or [HowTo100M](https://www.di.ens.fr/willow/research/howto100m/)? Such evaluations could better demonstrate your approach's practical applicability and robustness.

- *Introduction (Lines 38–40)**:  While your annotation approach is good ("gold-standard"), it could be beneficial to consider a scalable "silver-standard" dataset creation method where existing other datasets queries are generalized or rephrased automatically without direct human annotation, particularly relevant for instructional scenarios (e.g., generalizing "cutting tomato" to "cutting vegetables" in [YouCookII](http://youcook2.eecs.umich.edu/),).

- **Table 1**:  Important relevant datasets such as [YouCookII](http://youcook2.eecs.umich.edu/), [COIN](https://coin-dataset.github.io/), and [HiREST](https://github.com/j-min/HiREST) are missing. Including these would enhance completeness and better position your task relative to existing moment retrieval literature.

- **Related Work**:   Notable related methods such as [TMLGA](https://arxiv.org/abs/1908.07236), [DORi](https://arxiv.org/abs/2010.06260), and [ExCL](https://arxiv.org/abs/1904.02755) are absent from your literature review. Including these references would significantly enhance the completeness and clarity of your work’s context.

---

> ### Author Rebuttal · Authors · 2025-07-31
>
> We thank the reviewer for highlighting our practical multi-moment task and the effectiveness of the PV module. We address the concerns below:
>
> **W1, Q1:** We follow the bias evaluation methodology proposed in [1], particularly the analysis of temporal annotation bias in Fig. 2. We replicate similar experiments on QV-M² and our FlashMMR model.
>
> The results show that our dataset exhibits significantly lower temporal bias compared to other commonly used benchmarks such as Charades-STA[2] and TACoS[3]. As image uploads are not supported here, we report the distribution of normalized start/end positions in the training set below. The results demonstrate that QV-M² has a much more uniform temporal distribution, highlighting its advantage in mitigating temporal bias. "Pred." refers to the predictions generated by FlashMMR. "Interval" denotes the normalized temporal interval of the video.
>
> | Interval | QV-M² Start % | QV-M² End % | Charades-STA Start % | Charades-STA End % | TaCos Start % | TaCos End % | Pred. Start % | Pred. End % |
> |----------|---------|-------|---------|-------|---------|-------|---------|-------|
> | [0.0, 0.1) |19.90% | 5.32% |31.39% | 0.42% |21.82% | 11.53% |15.01% | 7.49% |
> | [0.1, 0.2) |8.73% | 8.02% |9.85% | 6.80% |16.47% | 14.71% |9.52% | 10.47% |
> | [0.2, 0.3) |9.84% | 10.05% |8.59% | 10.54% |12.61% | 13.44% |11.60% | 11.57% |
> | [0.3, 0.4) |9.04% | 9.06% |8.51% | 11.90% |9.90% | 9.45% |9.15% | 8.96% |
> | [0.4, 0.5) |10.26% | 10.34% |8.87% | 12.82% |7.92% | 8.96% |10.63% | 10.96% |
> | [0.5, 0.6) |9.87% | 10.15% |9.59% | 9.24% |7.19% | 6.44% |10.79% | 10.53% |
> | [0.6, 0.7) |9.08% | 8.56% |9.62% | 8.30% |6.77% | 7.58% |9.56% | 9.50% |
> | [0.7, 0.8) |8.49% | 8.57% |8.03% | 8.24% |5.40% | 5.32% |9.04% | 9.11% |
> | [0.8, 0.9) |9.54% | 9.99% |4.67% | 8.44% |5.26% | 6.55% |9.71% | 10.74% |
> | [0.9, 1.0) |5.25% | 19.96% |0.89% | 23.31% |6.66% | 16.03% |5.00% | 10.66% |
>
> &nbsp;
>
> **W2, Q2:** We thank the reviewer’s question regarding the computational overhead of our proposed Post-Verification module. To clarify, we conducted thorough experiments analyzing both runtime and scalability concerning video length and the number of candidate predictions.
>
> **Computational Complexity Analysis:**
> The theoretical complexity of the Post-Verification module are as follows:
>
> * **Inference:** Time Complexity: $O(K\bar{L}d^2)$, Memory Complexity: $O(K\bar{L}d)$
>
> * **Training:** Time Complexity: $O(K\bar{L}d^2 + \bar{V}^2d)$, Memory Complexity: $O(K\bar{L}d + \bar{V}^2)$
>
> Where, $K$ is the number of candidate predictions per video, $d$ is the feature dimension, $\bar{L}$ is the average length of candidate, $\bar{V}$ is the average length of video.
>
> **Experimental Validation:**
> We empirically validated the above theoretical analysis by varying video lengths and candidate counts. All experiments utilized a batch size of 32, a feature dimension of 256, and were performed on an NVIDIA RTX 4090 GPU with an Intel(R) Core(TM) i9-14900KF CPU.
>
> (1) Scalability with Video Length:
>
> | Video Length ($\bar{V}$) | Inference Runtime (s) | Inference FLOPs (M) | Training Runtime (s) | Training FLOPs (M) |
> |-----|---------|-------|--------|----------|
> | 50 | 0.0133 | 19051.2 | 0.0225 | 19175.0 |
> | 100 | 0.0132 | 19051.2 | 0.0229 | 19546.0 |
> | 150 | 0.0131 | 19051.2 | 0.0259 | 20164.4 |
> | 200 | 0.0130 | 19051.2 | 0.0340 | 21030.2 |
> | 250 | 0.0129 | 19051.2 | 0.0459 | 22143.3 |
> | 300 | 0.0130 | 19051.2 | 0.0598 | 23503.8 |
> | 400 | 0.0130 | 19051.2 | 0.0927 | 26966.8 |
>
> These results confirm our theoretical expectation: during inference, runtime and FLOPs remain stable irrespective of video length. In contrast, during training, runtime and FLOPs increase **linearly** with video length due to the $\bar{V}^2d$ term. Since $\bar{V}^2 < K\bar{L}d$ in our setting, the $\bar{V}^2d$ term remains a lower-order contribution, which explains why the additional FLOPs during training are moderate.
>
> (2) Scalability with Candidate Set Size:
>
> | Candidate Count ($K$) | Inference Runtime (s) | Inference FLOPS (M) | Training Runtime (s) | Training FLOPS (M) |
> |----|---------|--------|-------|-------|
> | 50 | 0.0134 | 19051.2 | 0.0261 | 20164.4 |
> | 60 | 0.0156 | 22861.5 | 0.0284 | 23974.7 |
> | 70 | 0.0181 | 26671.7 | 0.0310 | 27784.9 |
> | 80 | 0.0203 | 30482.0 | 0.0337 | 31595.2 |
> | 90 | 0.0230 | 34292.2 | 0.0364 | 35405.4 |
>
> These experiments demonstrate a clear **linear relationship** between candidate count and computational overhead in both inference and training phases, again aligning well with our complexity analysis.
>
> In summary, these experimental results demonstrate that, under the current setting, the computational complexity of our Post-Verification module is reasonable and acceptable.
>
> &nbsp;
>
> **W3:** The method indeed borrows several modules from previous approaches and combines them sensibly. However, these modules do not adapt well to the new MMR task. The newly proposed post verification module is a useful addition given the task setting, and is shown to work well in the ablation.
>
> &nbsp;
>
> **W4:** We would like to clarify that our MMR scenarios are not constructed by combining existing single-moment annotations. Instead, all multi-moment labels are manually annotated from scratch. Furthermore, to ensure annotation consistency and minimize potential bias, we employed a structured quality control process following the annotation phase.
>
> &nbsp;
>
> **C1:** Thank you for the suggestion. We will revise the abstract to emphasize that such cases already exist in previous datasets, in order to strengthen our motivation and clearly position the proposed MMR task.
>
> &nbsp;
>
> **C2:** Yes. We will clearly state that the new annotations explicitly include temporal sequencing information in the revised version.
>
> As mentioned in the second point of our annotation guidelines, both SMR and MMR annotations encode temporal sequencing information. For example, in the video *1tGsRVsMtlc_660.0_810.0*, the query *"A woman shows the clothing before pointing the camera at the mirror."* contains the temporal cue *"before"*, indicating that the event should occur prior to *"pointing the camera at the mirror."* If one were to simply align the visual content with the text while ignoring this temporal cue, the subsequent action would mistakenly be included, making it impossible to correctly determine the end time of the intended event.
>
> &nbsp;
>
> **C3, Q4:** Yes. We still evaluate FlashMMR on YouCookII (an instructional-video dataset for SMR), even though newly proposed PV module is meant to address the multi-moment retrieval (MMR) task.
>
> As a result, the single-moment retrieval (SMR) performance primarily reflects the capability of our baseline model, FlashVTG, as supported by the experimental results below. Our proposed module does not significantly affect the SMR task. Notably, videos in the YouCookII dataset are much longer than those in previously used datasets (e.g., QVHighlights and Charades-STA), while our baseline model is not well-suited for handling such long videos.
>
> |Method|G-mAP|mIoU@1|mR@1|
> |-|-|-|-|
> |FlashVTG (baseline)|12.50|17.87|13.38|
> |FlashMMR|12.45|17.57|13.04|
>
> &nbsp;
>
> **C4:** Thank you for the insightful suggestion. We agree that automatic query generalization, e.g., turning "cutting tomato" into "cutting vegetables", could help build a large "silver-standard" dataset.
>
> A practical caveat is that the generated queries from instructional datasets tend to stay at the action level, whereas mainstream MR benchmarks use event-level queries. Even so, the technique remains attractive for producing large-scale pre-training data. For the present work, we therefore focus on a manually curated, event-level benchmark and leave "silver-standard" expansion as promising future work.
>
> &nbsp;
>
> **C5:** Thank you for the suggestion. We have added additional relevant datasets (YouCookII, COIN, and HiREST) to Table 1 in the revised version.
>
> |Dataset|Domain|Avg #moment per query | Avg Query Len | Avg ratio Moment/Video | #Moments / #Videos | Fully Human-Annotated |
> |-|-|-|-|-|-|-|
> |YouCookII|Cooking|1|8.7|1.9%|13.8K/2K|✓|
> |COIN|Open|1|4.9|9.8%|46.3K/11.8K|✓|
> |HiREST|Open|1|4.2|55.0%|2.4K/526|✓|
>
> &nbsp;
>
> **C6:** Thank you for the valuable suggestion. We have included citations of TMLGA, DORi, and ExCL in the revised Related Work section, as they are highly relevant to our task.
>
> These methods all contribute to moment retrieval:
>
> - TMGLA proposes a proposal-free, end-to-end method for natural language-based temporal localization using a guided attention mechanism and soft labels to improve efficiency and accuracy.
>
> - DORi proposes a language-conditioned graph-based method for temporal moment localization. It models human-object-activity interactions via spatio-temporal graphs and uses iterative message passing to align video content with natural language queries.
>
> - ExCL proposes a modular framework that fuses video and text features via cross-modal LSTMs and directly predicts clip boundaries using span predictors.
>
> &nbsp;
>
> **Q3:** Yes. We agree that hierarchical or sparse interaction mechanisms may provide a more efficient way to model semantic consistency in the post-verification stage, especially by focusing on high-confidence segments.
>
> We adopt a GRU-based design with $L_{repr}$ to model local and global temporal consistency between predicted moments and the query. In contrast, purely hierarchical or sparse interaction mechanisms may lack fine-grained global context, and their supervision signals can become sparse. As noted in the computational complexity analysis, the design remains efficient in scenarios where $\bar{V}^2 < K\bar{L}d$. Nonetheless, we acknowledge the potential benefits of alternative interaction strategies, and we consider this an important direction for future exploration.
>
> &nbsp;
>
> [1] https://arxiv.org/abs/2101.09028
>
> [2] Tall: Temporal activity localization via language query.
>
> [3] Grounding action descriptions in videos.

---

> > ### Author Response · Authors · 2025-08-05
> > **Update for C4**
> >
> > **Update for C4:**
> >
> > Inspired by the **graph-based** design of DORi, which models human-object interactions conditioned on language to improve temporal moment localization, we see a promising future direction in developing a semi-automatic (**silver-standard**) pipeline to construct large-scale pretraining data for multi-moment retrieval.
> >
> > 1. First, we could reuse existing instruction-style annotations to segment videos into moments and use VLM to generate new captions for each moment. Make sure the caption is not limited to the action level.
> > 2. Next, we could perform scattered recurrence mining by parsing the captions into syntax trees, constructing **action scene graphs**, and identifying recurring entities across time.
> > 3. Based on these candidates, we could generate multi-moment query-timestamp pairs using LLMs.
> > 4. Finally, we could manually verify and refine both the queries and their associated time spans to ensure quality and alignment with the video content.
> >
> > We appreciate your constructive suggestions very much, and think such a semi-automatic pipeline could be a feasible way to construct large-scale pretraining data for multi-moment retrieval. What is your opinion?

---

### Note · Authors · 2025-08-13

We thank all reviewers constructive feedback and active discussions. We are glad to see most concerns resolved, with pD2D, T6Ks, and yR7Q expressing positive inclinations, and LJnf confirming that most concerns were addressed. Below we summarize key takeaways:

**1. Key Strength Summary**

1. Proposes a multi-moment retrieval (MMR) task that better reflects real-world scenarios, with **clear motivation and significance**. (pD2D, T6Ks, yR7Q)

2. QV-M² is the **first fully human-annotated MMR benchmark** with high-quality labels and a verification procedure, making it a solid dataset contribution. (yR7Q, LJnf, T6Ks)

3. FlashMMR achieves **strong, state-of-the-art performance** on MMR, establishing a competitive baseline. (pD2D, T6Ks, yR7Q, LJnf)

4. The **Post-Verification (PV) module** effectively models semantic consistency across moments, reduces redundancy, and is validated in ablations; the IoU-guided re-scoring is innovative. (pD2D, T6Ks, LJnf)

5. **Comprehensive experiments and ablations**, with cross-dataset validation supporting the benchmark’s effectiveness; implementation details are clearly provided. (pD2D, LJnf, yR7Q)

**2. Responses to Key Concerns**
|Reviewer|Concerns| Response|
|-|-|-|
|yR7Q|#1. Suggests evaluating temporal-aware video LLM on QV-M². #2. Discuss potential paths to expand MMR dataset scale. #3. Minor layout flaw.|Added VLM experiments on QV-M²(#1); discussed dataset expansion paths(#2); and fixed format(#3). Thanks for **strong support**(5. Accept).|
|pD2D|#1. Dataset temporal bias analysis. #2. Discussion on PV module computational overhead and scalability. #3. More related work. #4. Performance on instructional datasets.|Added temporal bias analysis(#1); detailed PV module computation analysis(#2); more related work(#3) and instructional dataset experiments(#4). Thanks for **raising the score**.|
|T6Ks|#1. Missing train/val/test sizes. #2. IoU thresholds for mR\@k. #3. Differences from QVHighlights. #4. The repeated use of the letter k.|Added dataset splits and IoU thresholds(#1 & #2); clarified differences from QVHighlights(#3); provided extra experiments to support the results. Thanks for the **positive rating**.|
|LJnf|#1. Minor layout flaw. #2. Framework figure inconsistent with code. #3. Insufficient detailed discussion on Dummy Encoder.|Corrected formatting flaws(#1) and clarified that ACA and the Dummy Encoder are not our contributions(#2 & #3). Glad to see **most of the concerns addressed** and thank you.|

---

### Decision · Program_Chairs · 2025-09-17

**Decision:**

Accept (poster)

**Comment:**

Final rating: 4. Borderline Accept/ 4: Borderline Accept/ 5: Accept/ 4: Borderline Accept. The paper introduces QV-M, a multi-moment extension of QVHighlights (2,212 annotations, 5,522 segments) with new metrics for multi-moment retrieval, and proposes FlashMMR—a post-verification framework using constrained temporal adjustment plus verification to prune low-confidence segments. Retraining six MR methods on QV-M and QVHighlights, it reports gains over prior SOTA on QV-M (+3.00 G-mAP, +2.70 mAP@3+tgt, +2.56 mR@3), establishing a stronger benchmark and baseline for realistic temporal grounding.

Reviewers agreed that the paper introduces an important, practical task that extends beyond traditional single-moment retrieval, and that the experimental results are solid, with strong quantitative and qualitative performance for multi-moment retrieval. However, they questioned that the MMVR datasets are adapted from existing ones—limiting originality and potentially introducing biases—and that the constructed multi-moment scenes may be unnatural. They also noted that explicitly modeling cross-moment interactions adds unquantified computational cost and relies largely on standard attention with limited novelty; that evaluation breadth and scalability are insufficient (no temporal-aware video-LLM baselines, limited dataset scale without concrete expansion plans); and that there are presentation/clarity issues, including a formatting error at line 223 (bold header/line break), a possible typo in “Post-Processing with Feature Refinement,” a confusing Figure 3 that is inconsistent with the code (dummy/text tokens, k/v usage), and scant detail on the dummy encoder.

After the rebuttal, reviewers indicated their concerns were satisfactorily addressed and leaned toward acceptance. The ACs concur, agreeing that the work defines a well-motivated multi-moment retrieval task, introduces QV-M²—the first fully human-annotated, verification-backed MMR benchmark—and shows FlashMMR achieving state-of-the-art results as a strong baseline. Therefore, the area chairs agree to accept the paper and recommend that the authors refine it according to the reviewers’ suggestions for the camera-ready version